# Effectiveness of the Ponseti Method in the Treatment of Clubfoot: A Systematic Review

**DOI:** 10.3390/ijerph20043714

**Published:** 2023-02-20

**Authors:** Elena López-Carrero, José Manuel Castillo-López, Miguel Medina-Alcantara, Gabriel Domínguez-Maldonado, Irene Garcia-Paya, Ana María Jiménez-Cebrián

**Affiliations:** 1Podiatry Clinical Area, University of Seville, c/ Avenzoar 6, 41009 Seville, Spain; 2Department Podiatry, Faculty of Nursing, Physiotherapy and Podiatry, University of Seville, c/Avenzoar 6, 41009 Seville, Spain; 3Department Nursing and Podiatry, Faculty of Health Sciences, University of Málaga, c/Arquitecto Francisco Peñalosa 3, Ampliación del Campus de Teatinos, 29071 Málaga, Spain; 4Instituto de Investigación Biomédica de Málaga (IBIMA), 29071 Malaga, Spain

**Keywords:** Ponseti method, clubfoot, pes equinovarus, effectiveness treatment

## Abstract

Clubfoot is a common congenital deformity of the lower limbs. It should be treated as soon as possible so that its correction is more easily achieved. The objective of this systematic review was to assess the effectiveness of the Ponseti method in the treatment of clubfoot. A bibliographic search was carried out in different databases, including PubMed and SciELO. Filters such as full text and randomized controlled trial were selected to find those articles that best matched our search. Among the results, we selected the ones that interested us, and the rest were discarded, either because they did not meet the requirements for our work or because they were repeated. In total, we collected 19 articles, but after using the critical evaluation instrument CASPe, 7 of them were eliminated, leaving us with a total of 12 articles for our systematic review. After analyzing the results obtained in the selected articles, we concluded that the Ponseti method is effective in the treatment of clubfoot, presenting a high success rate.

## 1. Introduction

Clubfoot is a common congenital deformity of the lower limbs. It is difficult to treat due to the pathological anatomy of the foot; therefore, it is important to understand the mechanism of correction and ensure patient follow-up [1]. This pathology affects musculoskeletal structures of the feet, including the cavus, varus, adductus and equinus [2]. The goal of treatment is to correct all components of the deformity through gradual ligamentous and muscular lengthening, achieving a flexible plantigrade foot without pain [3,4].

The incidence is 1 or 2 cases per 1000 children born. It is approximately three times more frequent in males than in females. Clubfoot is present unilaterally or bilaterally in 50% of cases [2,5,6,7,8]. Its etiology may be associated with myelodysplasia, arthrogryposis or multiple congenital deformities, but the most common presentation is in isolation, which is considered the idiopathic type [2]. If a clubfoot remains deformed for a long time or does not receive the correct treatment, changes in bone structures may appear. These changes will depend on the severity of the soft tissue contracture and the effects it would have on walking. Lack of treatment or improper treatment can cause psychological or functional damage [6,7].

In 1836, Guerin became known as the first physician to use plaster for clubfoot. In the 20th century, new technologies were developed to treat this condition, such as the Thomas device. In 1932, Kite proposed his conservative method based on gentle and repeated manipulations followed by plaster to immobilize, in an attempt to prevent forced and prolonged corrections. Finally, in the mid-1940s, Ignacio Ponseti, a Spanish doctor specializing in orthopedics, conducted several in-depth studies on the pathological and functional anatomy of clubfoot. After this, he perfected the method, later describing it in significant detail [5,7,8].

The beginning of the treatment is essential to obtain good results. Any foot deformity should be corrected as soon as possible to promote good balance [9]. In the first weeks of life, conservative treatment is recommended [1,3,7,10]. Although it has been shown that the age at which treatment begins makes no significant difference, a clubfoot can be corrected later on [11]. It has been demonstrated that the Ponseti method is the gold standard, being an effective and safe treatment for clubfoot. The method is based on gentle and repetitive manipulations that aim to stretch the soft tissues progressively, followed by cast immobilization weekly. First, it is important to identify the various musculoskeletal structures so that the manipulations are conducted correctly and accurately, and subsequently placing a cast that will be changed every week. Finally, the equinus deformity is corrected via performing Achilles tendon tenotomy with minimally invasive surgery, followed by immobilization for three weeks in a cast. After removing the postsurgical cast, an abduction brace is placed. In recent years, the Ponseti method has gained acceptance as a conservative way of approaching deformity and is widespread around the world. Using this technique, the rate of success is usually about 90%. Although relapses do occur, this represents a much lower percentage. One of the advantages of this method is the degree of mobility that it grants at the end of treatment. As a disadvantage, the success of the method depends on variables such as age, sex, early diagnosis, if the patient has associated deformities and the number of casts used [1,2,3,4,5,6,7,8,9,10,11,12,13,14,15].

The Pirani classification system is a valid and reproducible technique for evaluating clubfoot. It is used to measure the severity of the foot before starting, at check-ups and at the end of the treatment [3,5,6,13] (Table 1).

Dimeglio classification may also be used to assess clubfoot. It is based on forefoot and hindfoot flexibility and differentiates four types of feet: type I, flexible hindfoot and forefoot; type II, rigid hindfoot and flexible forefoot; type III, only the forefoot is rigid and type IV, both the hindfoot and forefoot are rigid [8].

Professionals must understand that not all families have the same socioeconomic level or the same capacities to understand what is being done to their child. In some cases, a stricter follow-up will be required to obtain the correct results [2,5,10,13,14].

The purpose of this study was to systematically review original studies to determine the efficacy of the Ponseti method in the treatment of clubfoot.

## 2. Materials and Methods

We conducted a search of PubMed and SciELO using the following search terms: “Ponseti method, clubfoot, pes equinovarus. Effectiveness treatment” (Table 2).

In PubMed, we introduced the “Ponseti method” keyword with the free full text filter and obtained 269 results. We then added the “Randomized controlled trial” filter, and 9 results appeared between the years 2007 and 2021. We chose 5 articles, and the rest were discarded.

In PubMed, we introduced the “clubfoot” keyword with the free full text filter and 832 results appeared. We then added the filter “randomized controlled trial”, and 11 results appeared between the years 2006 and 2021. As 1 of them was repeated, we were left with 6 articles, and the rest we discarded.

In SciELO, we introduced the keyword “clubfoot” without a filter, and we found 57 results, from which we chose 9 articles, and the rest were discarded.

In PubMed, we introduced the keyword “pes equinovarus” with the free full text filter and 815 results appeared. We added the filter “randomized controlled trial” and 11 results appeared between the years 2006 and 2021, of which 10 were repeated and 1 was discarded.

In SciELO, we introduced the “Ponseti method” keyword without a filter. We found 17 results, 6 of which were repeated. We chose 2 articles, and 10 were discarded.

In SciELO, we introduced the keyword “clubfoot” without a filter. We found 12 results and we did not choose any article.

In SciELO we introduced the keyword clubfoot without a filter. We found 5 results and we did not choose any article.

In PubMed we introduced the congenital keyword “talipes equinovarus” with the filter free full text and 849 results appeared. We added “randomized controlled trial” and 11 results appeared between the years 2006 and 2021. All the articles had already been previously reviewed, so we did not select any articles.

In total, 15 articles were chosen for systematic review. According to the critical evaluation instrument CASPe, 7 articles were excluded (Table 3). Figure 1 shows the PRISMA flow diagram. The articles chosen are as follows:Stress radiography in the assessment of residual deformity in clubfoot following postero-medial soft tissue release.Correcting congenital talipes equinovarus in children using three different corrective methods.Evaluación ultrasonográfica del tendón de Aquiles en niños con pie equino varo aducto congénito posterior a tenotomía de Aquiles con el método de Ponseti. Seguimiento a 12 semanas.Are scoring systems useful for predicting results of treatment for clubfoot using the Ponseti method?Ultrasonographic aspects of the Achilles tendon after tenotomy for the treatment of congenital clubfoot by the Ponseti technique.Does the presence of clubfoot delay the onset of walking?Ponseti method in Brazil: first ten years of a clubfoot website—users profile.

Figure 1 shows the PRISMA 2020 flow diagram for this systematic review, which included searches of database and registers only.

This study was authorized by PROSPERO under CRD42021270956.

Three co-authors reviewed (A.M., J.-C. and E.L.-C.) the study and one intervened when there was disagreement (I.G.-P.).

## 3. Results

Table 4 shows the results of the scientific studies organized by year. We can see the author, type of study, sample number and duration of study.

## 4. Discussion

In this review, we aimed to evaluate the efficacy of the Ponseti method for the treatment of clubfoot.

The goal is to obtain and maintain a flexible, plantigrade and pain-free foot. When clubfoot is not treated or is treated improperly, the affected child may suffer functional and psychological damage. It has great advantages, since in most cases a 90% success rate is achieved in patients. As those treated are usually young patients, from newborns to a few months of age, the tissues are very elastic, which favors correction through gentle manipulations with gradual ligamentous and muscular lengthening. The study by Sanghvi and Mittal, as well as the study by Lara et al., recommends that the technique be started in the first 15 days of life [1,8]. Alves et al. said that the age at which treatment is started makes no significant difference. The principles of the Ponseti technique are simple, and the health professional must have a thorough understanding of the deformity and be skilled with manipulations and cast changes [11]. On the other hand, we see some drawbacks, such as recurrence or the required use of an abduction orthosis to maintain the degrees of correction obtained. The latter can be uncomfortable for some children when the strict protocol of use is not complied with [2,5,7,12,14].

Authors comparing the techniques of Kite and Ponseti have found good results in both, but the Ponseti method seems to lead to better results than the Kite method in achieving clubfoot correction. With the Ponseti method, fewer casts were needed, and the duration of treatment was shorter. The maximum dorsiflexion achieved in the ankle was significantly greater, and residual deformity and recurrence were seen at slightly lower rates [1,7,10]. Gintautiene et al. compared Ponseti’s technique with an early transfer of the tibialis anterior tendon, with the latter allowing a reduction in the duration of the orthosis; however, a possible weakening of dorsiflexion was seen. The results obtained are the same as those seen with the Ponseti method; therefore, it is a good option to opt for the conservative treatment [15]. Zwick et al. compared surgical treatment with the Ponseti method. In this study, the patients treated with the Ponseti method obtained better results, with plantigrade feet and no pain. In addition, higher parental satisfaction and better mobility of the foot were achieved than with the other foot treatment [12].

The success of this technique depends on the manipulations and the regularity of cast changing, which should start as soon as possible. The casts should be placed from the toes to the groin, paying attention to circulation problems or skin rashes. An average of six casts are usually necessary, but this depends on the severity of the deformity that the foot presents. Studies have been carried out comparing different materials used in the Ponseti method. The classic material is plaster of Paris, which is a white, easy to mold and cheap plaster; another is semi-rigid synthetic soft cast, a lightweight material that can be applied and quickly withdrawn; and finally, there is semi-rigid fiberglass, which is lighter than the other materials but expensive. There were no significant differences between them in the healing process; however, some advantages and disadvantages were discovered when we evaluated the degree of satisfaction with the material [3,5,8,13].

When the foot has achieved correction of the cavus, varus and adductus deformities, but has a dorsiflexion of less than 10°, Achilles tendon tenotomy is indicated to correct the equine [14]. The foot is immobilized in a cast for three weeks. After removing the postsurgical cast, a Dennis Brown splint is placed on the corrected foot. The boots are placed at 45° of abduction with a 70° and 10° dorsiflexion external rotation. In the case of unilateral clubfoot, the normal side is held at 40–45° of external rotation. This orthosis must be worn 23 h/day during the first three months, leaving an hour free for the bathroom. The hours of use are reduced until it is used only at night and during naps. It is advised children wear the splint until 4 or 5 years of age to avoid relapses. Changulani et al. said that those children who do not use it will likely have a recurrence [1,2,4,8,9,12].

It has been shown in several studies that Pirani classification is very helpful throughout the treatment of clubfoot. It is based on three midfoot variables and three hindfoot variables. Each variable has a value from zero to one. Normally, a value is taken at the beginning and another value at the end of the study to check the progress of the foot. Dimeglio’s classification is also of great help and is based on the flexibility of the forefoot and hindfoot. In the study by Islam et al., they compared the scores of groups using both the Pirani and Dimeglio classifications. On the other hand, in the study by Aydin et al., figures are shown comparing the Pirani results at beginning of the Ponseti method and after Achilles tendon tenotomy, showing that there was an improvement in the patients [3,5,6,8,13].

It is important to inform parents and give them enough information so that they understand the medical procedure. It is essential they listen to professionals’ orders, make an appointment for the patient or call if there is a problem during any part of the process. It could be a great idea to supply leaflets in which we could explain the deformity and treatment [4]. On many occasions, a follow-up is necessary to see if the home treatment is achieved by families. Due to a lack of understanding on the subject, some families may abandon treatment or not understand the importance of treating the clubfoot on time. Some more common examples are those families who live in neighborhoods with a low socioeconomic status, low income and high poverty rates [2,5,8].

The study by Hui et al. shows us that the psychological well-being of parents of children with clubfoot can be affected by the treatment process, especially at the beginning of treatment, when the situation may worsen due to their limited experience with the handling or care of the first casts [3].

The study by Islam et al. makes a small improvement to the traditional Ponseti method. All the steps are the same except for the changing of casts, which is to be carried out twice a week. This reduces the time patients are immobilized in the cast. The accelerated Ponseti technique has been shown to have similar safety and is effective enough as a traditional treatment. This should be further investigated so that future generations have improved clubfoot correction [6].

As mentioned in the previous paragraph, we believe that further research is critical in assessing the cost and benefit obtained using the Ponseti method, especially in undeveloped countries with few resources [14].

The most important limitations of this study are that the number of clinical trials currently published is small, probably due to the few existing cases (1–2 children per 1000 are born with clubfoot according to the authors), and that the studies examined were clinical trials in children, which are difficult to conduct.

## 5. Conclusions

It is concluded that the Ponseti method is effective at an early age, with a success rate, according to the authors, of 90% in correcting clubfoot deformity, preventing from having to resort to surgery.

## Figures and Tables

**Figure 1 ijerph-20-03714-f001:**
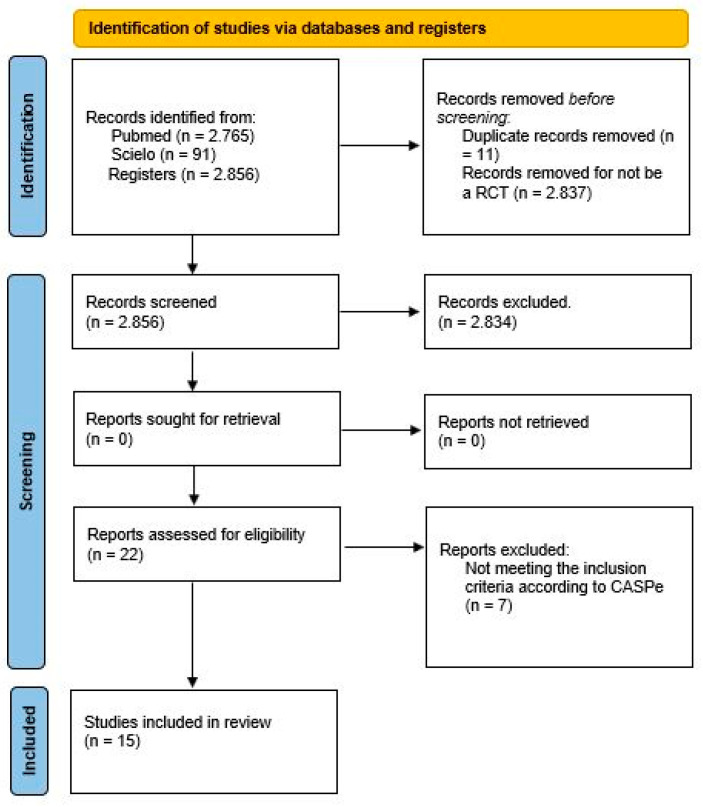
PRISMA diagram.

**Table 1 ijerph-20-03714-t001:** Pirani classification system [3] developed by Hui et al. 2014.

Pirani Classification System	
6 Clinical Signs are Measured as 0 (Normal), 0.5 (Moderately Normal) and 1 (Severe)	
MIDFOOT GRADUATION	HINDFOOT GRADUATION
There are 3 clinical signs	There are 3 clinical signs
(maximum 3 points)	(maximum 3 points)
CURVED SIDE EDGE	BACK FOLD
MEDIAL FOLD	RIGID EQUINE
TALUS HEAD COVERAGE	EMPTY HEEL

**Table 2 ijerph-20-03714-t002:** This figure shows the methodology followed.

Methodology					
PubMed	Ponseti method	Free full text 269 results	Randomized controlled trial	9 results	5 articles chosen
PubMed	Clubfoot	Free full text 832 results	Randomized controlled trial	11 results	6 articles chosen
SciELO	Clubfoot	Without filter		57 results	9 articles chosen
PubMed	Pes equinovarus	Free full text 815 results	Randomized controlled trial	11 results	0 articles chosen
SciELO	Ponseti Method	Without filter		17 results	2 articles chosen
SciELO	Pes equinovarus	Without filter		12 results	0 articles chosen
SciELO	Pie zambo	Without filter		5 results	0 articles chosen
PubMed	Congenital talipes equinovarus	Free full text 849 results	Randomized controlled trial	11 results	0 articles chosen

**Table 3 ijerph-20-03714-t003:** The critical evaluation instrument CASPe.

Title	CASPe Score	Results
Stress radiography in the assessment of residual deformity in clubfoot following postero-medial soft tissue release.	5/11	Eliminate
Treatment of idiopathic clubfoot using the Ponseti method.	10/11	Acceptable
Ponseti vs. Kite’s method in the treatment of clubfoot—a prospective randomised study.	10/11	Acceptable
Ponseti Method Does Age at the Beginning of Treatment Make a Difference?	9/11	Acceptable
Comparison of Ponseti versus surgical treatment for idiopathic clubfoot.	9/11	Acceptable
Early results of the Ponseti technique for a clubfoot clinic in South Africa.	9/11	Acceptable
Conservative management of idiopathic clubfoot: Kite versus Ponseti method.	10/11	Acceptable
Early results of treatment for congenital clubfoot using the Ponseti method.	9/11	Acceptable
Treatment of idiopathic congenital clubfoot using the Ponseti method: ten years of experience.	10/11	Acceptable
Comparison of cast materials for the treatment of congenital idiopathic clubfoot using the Ponseti method: a prospective randomized controlled trial.	10/11	Acceptable
Treatment of clubfoot with ponseti method using semirigid synthetic softcast.	11/11	Acceptable
Correcting congenital talipes equinovarus in children using three different corrective methods.	5/11	Eliminate
Functional and clinical results achieved in congenital clubfoot patients treated by Ponseti’s technique.	8/11	Acceptable
Comparison of the Ponseti method versus early tibialis anterior tendon transfer for idiopathic clubfoot: A prospective randomized study.	10/11	Acceptable
Treatment of congenital clubfoot using Ponseti method.	9/11	Acceptable
Evaluación ultrasonográfica del tendón de Aquiles en niños con pie equino varo aducto congénito posterior a tenotomía de Aquiles con el método de Ponseti. Seguimiento a 12 semanas.	4/11	Eliminate
Are scoring systems useful for predicting results of treatment for clubfoot using the Ponseti method?	5/11	Eliminate
Ultrasonographic aspects of the Achilles tendon after tenotomy for the treatment of congenital clubfoot by the Ponseti technique.	0/11	Eliminate
Evaluation of kite and Ponseti methods in the treatment of idiopathic congenital clubfoot.	10/11	Acceptable
Results of a standard versus an accelerated Ponseti protocol for clubfoot: a prospective randomized study.	11/11	Acceptable
Does the presence of clubfoot delay the onset of walking?	5/11	Eliminate
Ponseti method in Brazil: first ten years of a clubfoot website—users profile.	0/11	Eliminate

**Table 4 ijerph-20-03714-t004:** Study results.

Title	Author	Year	Type of Study	Sample (N)	Duration of Study	Results
Treatment of idiopathic clubfoot using the Ponseti method	Changulani et al. [4]	2006	Randomized controlled trial	66 patients (100 clubfoot)	2 years	The use of the Ponseti method, which is a simple and effective method to treat congenital idiopathic clubfoot, led to the prevention of surgery in up to 89% of cases.
Ponseti vs. Kite’s method in the treatment of clubfoot—a prospective randomised study	Sud et al. [10]	2008	Randomized controlled trial	45 patients (67 clubfoot)	1 year	It was found that the correction of clubfoot was significantly improved with the Ponseti method (with fewer days and fewer casts) as compared to the Kite method.
Ponseti Method Does Age at the Beginning of Treatment Make a Difference?	Alves et al. [11]	2009	Randomized controlled trial	68 patients (102 clubfoot)	2 years	All feet (100%) were initially corrected and no feet required extensive surgery, regardless of age, at start of treatment. There were no differences between Groups I and II in the number of casts, tenotomies, success in terms of rate of initial correction, recurrence rate and tibial transfer rate previous. The rate of extensive surgery avoidance when using the Ponseti method was 100% in Groups I and II; relapses occurred in 8% of feet in young and older children.
Comparison of Ponseti versus Surgical treatment for idiopathic clubfoot	Zwick et al. [12]	2009	Randomized controlled trial	19 patients (28 clubfoot)	3 years and a half	In the group for which the Ponseti method was performed, better results were obtained; no patient had foot pain or plantigrade feet.
Early results of the Ponseti technique for a clubfoot clinic in South Africa	Firth et al. [14]	2009	Randomized controlled trial	70 patients (106 clubfoot)	6 years	The Ponseti technique led to good results for clubfoot, but no significant differences were found between the group that received previous treatment and the group that did not.
Conservative management of idiopathic clubfoot: Kite versus Ponseti method	Sanghvi y Mittal [1]	2009	Randomized controlled trial	42 patients (64 clubfoot)	3 years	The results obtained with the Kite and Ponseti methods were similar (79% vs. 87%). With the Ponseti method, the number of casts was significantly lower, as well as the time needed to achieve correction, and a greater maximum ankle dorsiflexion was obtained.
Early results of treatment for congenital clubfoot using the Ponseti method	Matuszewski et al. [9]	2011	Randomized controlled trial	35 patients (47 clubfoot)	4 years	All patients achieved satisfactory results, and there have not been any known recurrences.
Treatment of idiopathic congenital clubfoot using the Ponseti method: ten years of experience	Lara et al. [8]	2013	Randomized controlled trial	155 patients (229 clubfoot)	10 years	The Ponseti method was used in both groups and correction was achieved, but in the group with younger patients, the results were more satisfactory and fewer casts were used.
Comparison of cast materials for the treatment of congenital idiopathic clubfoot using the Ponseti method: a prospective randomized controlled trial	Hui et al. [3]	2014	Randomized controlled trial	30 patients (44 clubfoot)	2 years and a half	When performing the Ponseti method, there were no significant differences when using different materials in the mean number of casts needed to correct clubfoot.
Treatment of clubfoot with Ponseti method using semirigid synthetic softcast	Aydin et al. [13]	2015	Randomized controlled trial	196 patients (249 clubfoot)	1 year	There were no significant differences in the results of the Ponseti method when using the classical material (POP) or using a semirigid synthetic soft cast.
Functional and clinical results achieved in congenital clubfoot patients treated by Ponseti’s technique	Jaqueto et al. [5]	2016	Randomized controlled trial	31 patients (51 clubfoot)	5 years and a half	The Ponseti method offered functional and clinical efficacy in patients with a success rate of 90.2% and an improvement in the Pirani classification values.
Comparison of the Ponseti method versus early tibialis anterior tendon transfer for idiopathic clubfoot: A prospective randomized study	Gintautiene et al. [15]	2016	Randomized controlled trial	39 patients (55 clubfoot)	2 years	An early transfer of the tibialis anterior tendon allowed a reduction in the duration of the use of the orthosis and led to the same results as with the Ponseti method. However, a less significant difference in dorsiflexion of the foot was observed in those patients who underwent a tibialis anterior tendon transfer.
Treatment of congenital clubfoot using Ponseti method	Chueire et al. [2]	2016	Randomized controlled trial	26 patients (39 clubfoot)	4 years	The Ponseti method offered good clubfoot results with fewer soft tissue injuries.
Evaluation of Kite and Ponseti methods in the treatment of idiopathic congenital clubfoot	García et al. [7]	2018	Randomized controlled trial	100 patients (127 clubfoot)	1 year and 4 months	It was shown that the efficacy of the treatment with the Ponseti method was higher than with the Kite method.
Results of a standard versus an accelerated Ponseti protocol for clubfoot: A prospective randomized study	Islam et al. [6]	2020	Randomized controlled trial	100 patients (158 clubfoot)	1 year	The data suggest that performing the accelerated Ponseti technique, that is, changing casts twice a week, reduces immobilization time without affecting the final results and is as safe and effective as the traditional Ponseti method.

## Data Availability

Not applicable.

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
