# Peer review of "Effectiveness of the Ponseti Method in the Treatment of Clubfoot: A Systematic Review"

_ijerph, 2023, doi:10.3390/ijerph20043714_

Round 1

Reviewer 1 Report (Previous Reviewer 1)

The article reads better after the revision; however a few more grammatical changes should be made before publication:

1. Line 47: change plasters to plaster 

2. Lines 66-77.  Would eliminate the brand names of the braces and simply list as an abduction brace

Line 75: remove "developed by Dr. Pirani" as it is redundant since his named classifications is listed in the preceding few words.

Line 89/90: You should list all of the search terms.  For example on line 102 you state that "pes equinovarus" was used as a search term but don't mention it above.

Table 2: Justification of the columns is off.  I think its related to the Pubmed rows having 5 columns, but the SciELO rows only having 4.   You should be uniform and if something doesn't apply list a hash mark. 

Discussion line 36.  Should read "The classic material is plaster of Paris (POP).  If the abbreviation isn't used in the remainder of the manuscript you also don't need to abbreviate.

Author Response

This manuscript is a resubmission of an earlier submission. The following is a list of the peer review reports and author responses from that submission.

Round 1

Reviewer 1 Report

While the review is comprehensive in nature, there are numerous grammatical and stylistic errors that should be corrected prior to publication.  For example:

line 33: should be adductus and equinus

Line 35 would insert flexible before plantigrade and eliminate "and good mobility"

Line 43: would change "done in an inadequate way can" to "treated inadequately"

Lines 45-49.  While a historical overview is relevant, I'm not sure you have to include remote history in the time of Hippocrates.

Line 56-57: eliminate "and aid in learning to walk

Line 59: not a complete sentence 

Line 60: Gold does not need to be capitalized.  Would add "the" before.  

Line 60: "being..clubfoot" is awkward phrasing.

Line 61: would change "his" to "the"

Line 63.  eliminate the sentence started with "First"

Line 67: would take out the proper name of the splint "Dennis Brown" as many people don't use that.  I personally use a Dobbs brace

Line 68,69: awkward phrasing

Line 79: eliminate "allowing to keep a control"

Line 82: the Dimeglio classification is another common classification that should be mentioned in the review

Line 86: would eliminate the sentence "the family..."

Methods section: Can condense to something like "A search of Pubmed and Scielo using the following search terms: "Ponseti method, clubfoot..."  Good use of flow diagram.  Also would take out and just replace with a sentence saying the articles were excluded.

discussion: 

Line 2: awkward phrasing

Line 18: would take out the sentence that begins "it is curious"

Line 32: would take out plaster and change to cast as many people use semi-soft fiberglass instead of plaster

Line 36: should define POP

line 42: should be adductus

Line 49: "little by little" not appropriate phrasing. 

Line 76: would take out "seems very interesting to us"

Line 85: limitations paragraph should be rewritten

conclusions: should be re-written with better phrasing

Reviewer 2 Report

The topic of the article "Effectiveness of the Ponseti Method in the Treatment of Club" is in my opinion very important and interesting. I agree with the messages provided by the authors in the introduction and discussion. I agree with the conclusions presented by the authors.

Unfortunately, I completely disagree with the methodology.

It is obvious to me why the authors used the PubMed database for the analysis. I fully agree with that.

I don't understand why Scielo was used.

As the authors write "Ignacio Ponseti, (was) a Spanish doctor specializing in orthopedics, conducted several in-depth studies on the pathological and functional anatomy of clubfoot. After this, he contributed and perfected the method describing many details", with which I agree . I confirm the great influence of Spanish orthopedists in the development of this scientific problem.

This, however, does not explain why one of the two, few databases used is a database used in the Spanish-speaking cultural zone, practically not used elsewhere.

There are many other databases that are much more common.

Second problem. I understand why the authors chose a randomized controlled trial. However, it is not acceptable for me to choose the free full text option.

Many valuable articles are always included among those that do not have the free full text option.

The inclusion of Scielo and not other international medical databases and the positive free full text option is a problem in my opinion.

Maybe formally it is "a systematic review", but in my opinion the article is not in line with the idea of this type of publication.

Since the article seems interesting and important to me, I suggest publishing it as a review, not a "systematic review".

Other options I suggest:

- supplementing the article with other international databases + removing the free full text option;

- entering in the title +/- "experience of Spanish orthopedists" + marking in the text that it is logical, because Ignacio Ponseti was Spanish and the authors are his scientific heirs.

I apologize for my critical remarks, because I like the article in terms of the thoughts conveyed. In addition, I use the method of Ignacio Ponseti myself in the treatment of patients.

I hope we will find a consensus. I think the simplest change would be to change the title.
